# Endothelial PKA activity regulates angiogenesis by limiting autophagy through phosphorylation of ATG16L1

Xiaocheng Zhao[1,2], Pavel Nedvetsky[1,2,3], Fabio Stanchi[1,2], Anne-Clemence Vion[4,5], Oliver Popp[6], Kerstin Zühlke[7], Gunnar Dittmar[6,8], Enno Klussmann[7,9], Holger Gerhardt[1,2,4,9,10]*

[1]Vascular Patterning Laboratory, Center for Cancer Biology, VIB, Leuven, Belgium; [2]Vascular Patterning Laboratory, Center for Cancer Biology, Department of Oncology, VIB, Leuven, Belgium; [3]Medical Cell Biology, Medical Clinic D, University Hospital Münster, Münster, Germany; [4]Integrative Vascular Biology Lab, Max-Delbrück Center for Molecular Medicine in the Helmholtz Association (MDC), Berlin, Germany; [5]INSERM UMR-970, Paris Cardiovascular Research Center, Paris Descartes University, Paris, France; [6]Proteomics, Max-Delbrück Center for Molecular Medicine in the Helmholtz Association (MDC), Berlin, Germany; [7]Anchored Signaling Lab, Max-Delbrück Center for Molecular Medicine in the Helmholtz Association (MDC), Berlin, Germany; [8]CRP Santé · Department of Oncology, LIH Luxembourg Institute of Health, Luxembourg, Luxembourg; [9]DZHK (German Center for Cardiovascular Research), Berlin, Germany; [10]Berlin Institute of Health (BIH), Berlin, Germany

*For correspondence:
holger.gerhardt@mdc-berlin.de

**Abstract** The cAMP-dependent protein kinase A (PKA) regulates various cellular functions in health and disease. In endothelial cells PKA activity promotes vessel maturation and limits tip cell formation. Here, we used a chemical genetic screen to identify endothelial-specific direct substrates of PKA in human umbilical vein endothelial cells (HUVEC) that may mediate these effects. Amongst several candidates, we identified ATG16L1, a regulator of autophagy, as novel target of PKA. Biochemical validation, mass spectrometry and peptide spot arrays revealed that PKA phosphorylates ATG16L1α at Ser268 and ATG16L1β at Ser269, driving phosphorylation-dependent degradation of ATG16L1 protein. Reducing PKA activity increased ATG16L1 protein levels and endothelial autophagy. Mouse in vivo genetics and pharmacological experiments demonstrated that autophagy inhibition partially rescues vascular hypersprouting caused by PKA deficiency. Together these results indicate that endothelial PKA activity mediates a critical switch from active sprouting to quiescence in part through phosphorylation of ATG16L1, which in turn reduces endothelial autophagy.

## Introduction

Angiogenesis is the process of new blood vessels formation from pre-existing vessels via sprouting and remodeling. Blood vessels are crucial for tissue growth and physiology in vertebrates since they are the pipelines for oxygen and nutrients supply and for immune cell distribution. Inadequate vessel formation and maintenance as well as abnormal vascular remodeling cause, accompany or aggravate many disease processes including myocardial infarction, stroke, cancer, and inflammatory disorders (*Geudens and Gerhardt, 2011*; *Potente et al., 2011*).

Sprouting angiogenesis is a multistep process encompassing sprout initiation, elongation, anastomosis and final vascular network formation (*Geudens and Gerhardt, 2011*). Multiple molecular pathways have been identified to regulate sprouting angiogenesis; the most investigated are vascular endothelial growth factor (VEGF) and Notch/delta-like 4 (DLL4) signaling. Whereas VEGF initiates angiogenesis through activating VEGF receptor 2 (VEGFR2), thereby guiding angiogenic sprouting and tip cell formation as well as stalk cell proliferation (*Ferrara et al., 2003*; *Gerhardt et al., 2003*), Notch and its ligand delta-like 4 (DLL4) limit tip cell formation and angiogenic sprouting (*Hellström et al., 2007*; *Leslie et al., 2007*; *Lobov et al., 2007*; *Suchting et al., 2007*). VEGF and Notch/DLL4 pathways co-operate to form a fine-tuned feedback loop to balance tip and stalk cells during developmental angiogenesis and to maintain vascular stabilization (*Hellström et al., 2007*; *Leslie et al., 2007*; *Lobov et al., 2007*; *Suchting et al., 2007*). In addition, PI3K/Akt (*Lee et al., 2014*), MAP4K4 (*Vitorino et al., 2015*), ephrins and Eph receptors (*Cheng et al., 2002*), hedgehog signaling (*Pola et al., 2001*),YAP/TAZ (*Kim et al., 2017*; *Neto et al., 2018*),BMP signaling (*Vion et al., 2018*) and other pathways also regulate various aspects of angiogenesis.

We identified endothelial cAMP-dependent protein kinase A (PKA) as a critical regulator of angiogenesis during vascular development in vivo. Inhibition of endothelial PKA results in hypersprouting and increased numbers of tip cells, indicating that PKA regulates the transition from sprouting to quiescent vessels (*Nedvetsky et al., 2016*). However, the PKA targets mediating these effects remained unknown.

Here, we established a chemical genetics approach based on the mutation of a 'gatekeeper residue' of the ATP-binding pocket of a protein kinase to identify endothelial-specific substrates of PKA. We verified autophagy-related-protein-16-like 1 (ATG16L1) as a substrate of endothelial PKA and identified that phosphorylation of ATG16L1 facilitates its degradation. in vivo experiments showed that autophagy inhibition partially rescued the hypersprouting and increased tip cell numbers caused by PKA deficiency.

## Results

### Screen for novel substrates of endothelial PKA

In order to identify direct substrates of endothelial PKA, we employed a chemical genetics approach (*Allen et al., 2005*). The ATP-binding pocket of kinases contains a conserved 'gatekeeper residue', which in naturally occurring wild type (WT) kinases is usually a methionine or phenylalanine. In engineered analogue specific (AS) kinases, this 'gatekeeper residue' is replaced with a smaller amino acid (glycine or alanine), enabling the AS-kinases to accept ATP analogues (or ATPγS analogues) that are modified at the $N^6$ position with bulky groups as (thio-)phosphodonors. In contrast, WT-kinases poorly use these analogues. Once the substrates are thiophosphorylated by AS-kinases, they can be further alkylated and therefore recognized by a thiophosphate ester-specific antibody (*Alaimo et al., 2001*; *Allen et al., 2005*; *Allen et al., 2007*; *Banko et al., 2011*). To generate AS-PKACα, we mutated the methionine 120 to a glycine residue. Testing seven different variants of $N^6$-substituted bulky ATPγS analogues, we identified 6-cHe-ATPγS as the best thiophosphodonor for AS-PKACα substrates in HUVEC lysates (*Figure 1—figure supplement 1*).

To identify endothelial substrates of PKACα, HUVECs expressing WT-PKACα or AS-PKACα were lysed in kinase lysis buffer (KLB) and the thiophosphorylation reaction with 6-cHe-ATPγS was performed. After alkylation with p-Nitrobenzyl mesylate (PNBM), thiophosphorylated proteins were immunoprecipitated with the thioP antibody coupled to rProtein G Agarose beads (*Figure 1A*). For quality control, one thirtieth of the protein on agarose beads was eluted for western blot analysis, and the same amount of protein was used for gel silver staining (*Figure 1B*). The rest was subjected to mass spectrometry analysis. Two independent experiments were performed. Candidate endothelial PKA targets were identified as peptides that were at least 2-fold (log ratio(AS/WT)>1) enriched in the AS-PKACα samples compared to WT- PKACα samples in both experiments.

The ninety-seven proteins identified in both experiments (*Table 1—source data 1*) included several known PKA substrates such as NFATC1 (*Niswender et al., 2002*), VASP (*Anton et al., 2014*; *Butt et al., 1994*; *Profirovic et al., 2005*), PRKAR2A and PRKAR2B (*Manni et al., 2008*) indicating that the chemical genetic screen worked well to identify PKA substrates. Thirty proteins with at least

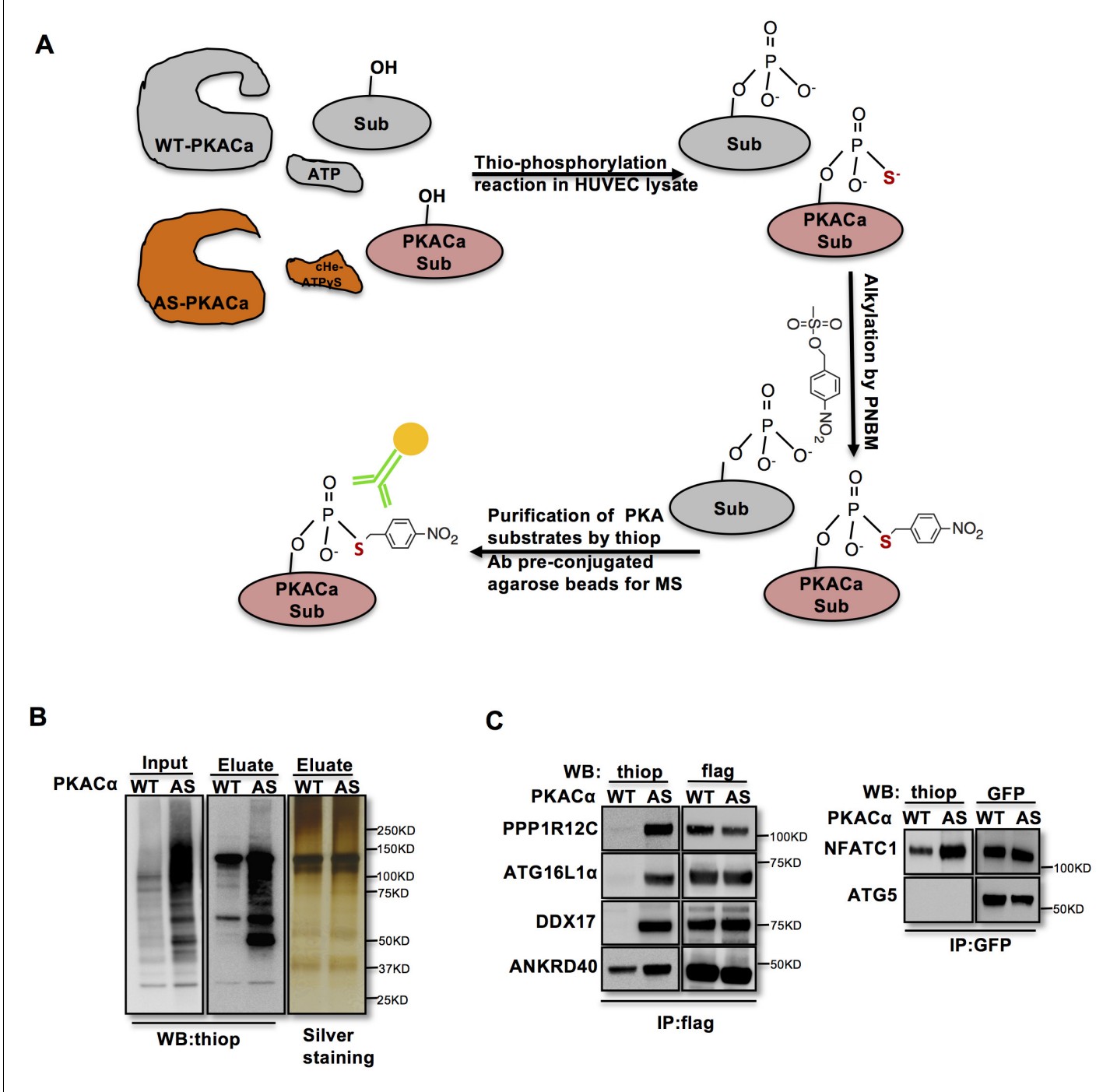

**Figure 1.** Identification of direct substrates of PKACα. (**A**) Strategy for labeling, immunoprecipitation and identifying of PKACα substrates in HUVEC lysates. (**B**) Thio-phosphorylation of PKA substrates in HUVEC lysates expressing WT-PKACα or AS-PKACα for mass spectrometry analysis. Left panel (input) shows western blot analysis of lysates after alkylation before immunoprecipitation, middle panel (Eluate) shows western blot analysis of the eluted proteins from the immunoprecipitation beads, right panel (Eluate) shows the silver staining of the same samples as middle panel. (**C**) Validation of the six PKACα substrates identified in the chemical genetic approach screen by overexpressing of potential substrates and WT-PKACα (or AS-PKACα) in 293Tcells, and labeling the substrate in 293 T cell lysates. Western blots are representative of two independent experiments.

The online version of this article includes the following figure supplement(s) for figure 1:

**Figure supplement 1.** Screen for the best N[6]-substituted ATPγS analog as a phosphodonor of AS-PKACα.

8-fold enrichment (log ratio(AS/WT)>3) are listed according to their log ratio (AS/WT) value in *Table 1*, presenting the most likely direct substrates of PKA in this screen.

To validate novel candidate PKACα substrates, we overexpressed WT-PKACα or AS-PKACα together with flag- or GFP- tagged candidate proteins in 293 T cells, and used the 6-cHe-ATPγS as thiophosphate donor to thiophosphorylate substrates in lysates as described above. Lysate

**Table 1.** List of PKACa substrates.

Proteins are listed according to the log of fold changes of AS-PKACα to WT- PKACα. Two independent experiments have been done to prepare the PKACα substrates samples for mass spectrometric analysis.

| | | | | Log ratio AS/WT | |
|---|---|---|---|---|---|
| Uniprot | Protein.names | Gene.names | Peptides | experiment1 | experiment2 |
| Q6AI12 | Ankyrin repeat domain-containing protein 40 | ANKRD40 | 9 | 10 | 10 |
| Q6P6C2 | RNA demethylase ALKBH5 | ALKBH5 | 6 | 10 | 10 |
| Q9NRY4 | Rho GTPase-activating protein 35 | ARHGAP35 | 9 | 10 | 10 |
| E7EVC7 | Autophagy-related protein 16–1 | ATG16L1 | 8 | 10 | 10 |
| J3KPC8 | Serine/threonine-protein kinase SIK3 | SIK3;KIAA0999 | 5 | 10 | 10 |
| A1 × 283 | SH3 and PX domain-containing protein 2B | SH3PXD2B | 4 | 10 | 10 |
| Q8IWZ8 | SURP and G-patch domain-containing protein 1 | SUGP1 | 5 | 10 | 10 |
| Q9UJX5 | Anaphase-promoting complex subunit 4 | ANAPC4 | 5 | 10 | 10 |
| O43719 | HIV Tat-specific factor 1 | HTATSF1 | 4 | 10 | 10 |
| O95644-5 | Nuclear factor of activated T-cells, cytoplasmic 1 | NFATC1 | 5 | 10 | 10 |
| G8JLI6 | Prolyl 3-hydroxylase 3 | LEPREL2 | 3 | 10 | 10 |
| F8W781 | Zinc finger CCCH domain-containing protein 13 | ZC3H13 | 3 | 10 | 10 |
| Q9BZL4 | Protein phosphatase 1 regulatory subunit 12C | PPP1R12C | 21 | 6,04440274 | 7,30701515 |
| O14974 | Protein phosphatase 1 regulatory subunit 12A | PPP1R12A | 26 | 5,72796034 | 7,12654716 |
| Q00537 | Cyclin-dependent kinase 17 | CDK17 | 31 | 6,37867381 | 6,39216838 |
| Q9Y4G8 | Rap guanine nucleotide exchange factor 2 | RAPGEF2 | 21 | 10 | 6,17455504 |
| Q9BYB0 | SH3 and multiple ankyrin repeat domains protein 3 | SHANK3 | 32 | 5,26591421 | 5,6389181 |
| J3KSW8 | Myosin phosphatase Rho-interacting protein | MPRIP | 18 | 4,61398477 | 5,61155414 |
| P31323 | cAMP-dependent protein kinase type II-beta regulatory subunit | PRKAR2B | 19 | 7,05077105 | 5,33509437 |
| P13861 | cAMP-dependent protein kinase type II-alpha regulatory subunit | PRKAR2A | 24 | 5,42841998 | 5,04010629 |
| Q14980-2 | Nuclear mitotic apparatus protein 1 | NUMA1 | 61 | 3,45625969 | 4,47466712 |
| O15056 | Synaptojanin-2 | SYNJ2 | 13 | 4,64022655 | 4,46069701 |
| J3KNX9 | Unconventional myosin-XVIIIa | MYO18A | 10 | 10 | 4,43208178 |
| Q86UU1-2 | Pleckstrin homology-like domain family B member 1 | PHLDB1 | 19 | 5,4105243 | 4,10782285 |
| P28715 | DNA repair protein complementing XP-G cells | ERCC5;BIVM-ERCC5 | 8 | 3,3571826 | 4,09305592 |
| P12270 | Nucleoprotein TPR | TPR | 104 | 3,3333472 | 4,04477536 |
| Q15111 | Inactive phospholipase C-like protein 1; Phosphoinositide phospholipase C | PLCL1 | 10 | 10 | 3,32188704 |
| Q9HD67 | Unconventional myosin-X | MYO10 | 39 | 4,13973415 | 3,21827463 |
| Q14185 | Dedicator of cytokinesis protein 1 | DOCK1 | 34 | 4,50413426 | 3,18515106 |
| O75116 | Rho-associated protein kinase 2 | ROCK2 | 29 | 3,27701864 | 3,08277835 |

The online version of this article includes the following source data for Table 1:
Source data 1. The full list of proteins identified in both experiments is provided.

immunoprecipitation was carried out with M2 anti-flag beads or anti-GFP antibody coupled agarose beads, and the immune complexes were probed by western blot using thiophosphate antibody. The known PKA substrate NFATC1 served as a positive control. Five selected new candidate proteins (PPP1R12C, ATG16L1α, DDX17, ANKRD40 and ATG5) out of ninety-seven proteins were tested; four of these five proteins were confirmed to be thiophosphorylated by AS-PKACα, indicating that they are indeed direct substrates of AS-PKACα (*Figure 1C*). Only ATG5 was not thiophosphorylated by AS-PKACα in the validation of the screen (*Figure 1C*). Bioinformatic analysis of ATG5 amino acids sequence also failed to identify a consensus PKA substrate motif (R-R/K-X-S/T;K/R-X$_{1-2}$-S/T) (*Kennelly and Krebs, 1991*). Since ATG5 directly binds to ATG16L1 (*Matsushita et al., 2007*; *Mizushima et al., 1999*), it likely co-precipitated with ATG16L1 in our screen.

ATG5 and ATG16L1 are conserved core components of the autophagy process, and PKA activity has been shown to negatively regulate autophagy in S. Cervisiae and mammalian cells through phosphorylation of ATG1/ULK1 (*Mizushima, 2010*). ATG16L1 however has not previously been identified as a PKA target, prompting us to further investigate this interaction and the potential regulatory role of PKA and autophagy in endothelial sprouting.

## Pkacα phosphorylates ATG16L1α at S268 and ATG16L1β at S269

To identify the PKACα phosphorylation sites in ATG16L1, we spot-synthesized 25-mer overlapping peptides that cover the entire ATG16L1 protein. The peptide array was subjected to an in vitro PKA phosphorylation assay. Three peptides of the ATG16L1α and 7 peptides of ATG16L1β were phosphorylated compared to the negative control (*Figure 2A*, *Supplementary file 1*). The common amino acid sequences included in the phosphorylated peptides predicted Ser268 in ATG16L1α and Ser269 as well as Ser287 in ATG16L1β as potential PKA phosphorylation sites (*Figure 2B*). Indeed serine to alanine mutation S268A in ATG16L1α and S269A but not S287A mutation in ATG16L1β resulted in a loss of AS-PKACα thiophosphorylation of these two ATG16L1 isoforms (*Figure 2C and D*). Moreover, LC-MS/MS analysis demonstrated phosphorylation of ATG16L1α at S268 and of ATG16L1β at S269 (*Figure 2E and F*). Of note, thiophosphorylation was converted to normal phosphorylation by 1% TFA acid-catalyzed hydrolysis during sample preparation for LC-MS/MS, thus identifying the target sites as phosphorylated, not thiophosphorylated (*Figure 2—figure supplement 1*). Together, these results demonstrate that S268 and S269 are the PKACα phosphorylation sites of ATG16L1α and ATG16L1β, respectively.

## Pkacα regulates ATG16L1 by phosphorylation-dependent degradation

Phosphorylation is one of the most widespread types of post-translational modification, and is crucial for signal transduction (*Hunter, 1995*; *Manning et al., 2002*; *Ubersax and Ferrell, 2007*). Previous research demonstrated that phosphorylation can regulate protein degradation by controlling its stabilization (*Bullen et al., 2016*; *Geng et al., 2009*; *Hwang et al., 2009*). To determine whether phosphorylation of ATG16L1α by PKA regulates protein stability, we overexpressed ATG16L1α$^{WT}$ and the mutant ATG16L1α$^{S268A}$ in HUVECs. To activate PKA, HUVECs were treated with the PKA specific activator 6-Bnz-cAMP. Using cycloheximide (CHX) to prevent new protein synthesis allowed us to detect the degradation of ATG16L1α$^{WT}$ and ATG16L1α$^{S268A}$ over time. Western blot analysis showed that most of the ATG16L1α$^{WT}$ degraded after 12 hr, whereas ATG16L1α$^{S268A}$ remained largely stable (*Figure 3A*), suggesting that the phosphorylation of ATG16L1α at site Ser268 by PKA promotes degradation. For ATG16L1β, Ser269 phosphorylation exhibited a similar function (*Figure 3B*). In addition, the phosphomimetic site mutants ATG16L1α$^{S268D}$ and ATG16L1β$^{S269D}$ were less stable than wild type ATG16L1α and ATG16L1β (*Figure 3C–3D*), supporting the idea that phosphorylation of ATG16L1α on site S268 and ATG16L1β on site S269 by PKA promotes degradation. In accordance, depleting PKACα in HUVECs by shRNA led to accumulation of ATG16L1 (*Figure 3E*) whereas activating PKA by 6-Bnz-cAMP caused ATG16L1 reduction (*Figure 3F*). Moreover, in PKA inhibited endothelial cells isolated from dnPKA$^{iEC}$ mice (*Nedvetsky et al., 2016*), ATG16L1 protein was also increased compared to endothelial cells isolated from corresponding Cdh5-CreERT2 control mice (*Figure 3G*). dnPKA$^{iEC}$ is the short denomination for Prkar1a$^{Tg/+}$ mice carrying a single floxed dominant-negative Prkar1a allele, (the regulatory subunit Prkar1a of PKA is an endogenous inhibitor of PKA) crossed with Cdh5-CreERT2 mice expressing tamoxifen inducible Cre recombinase

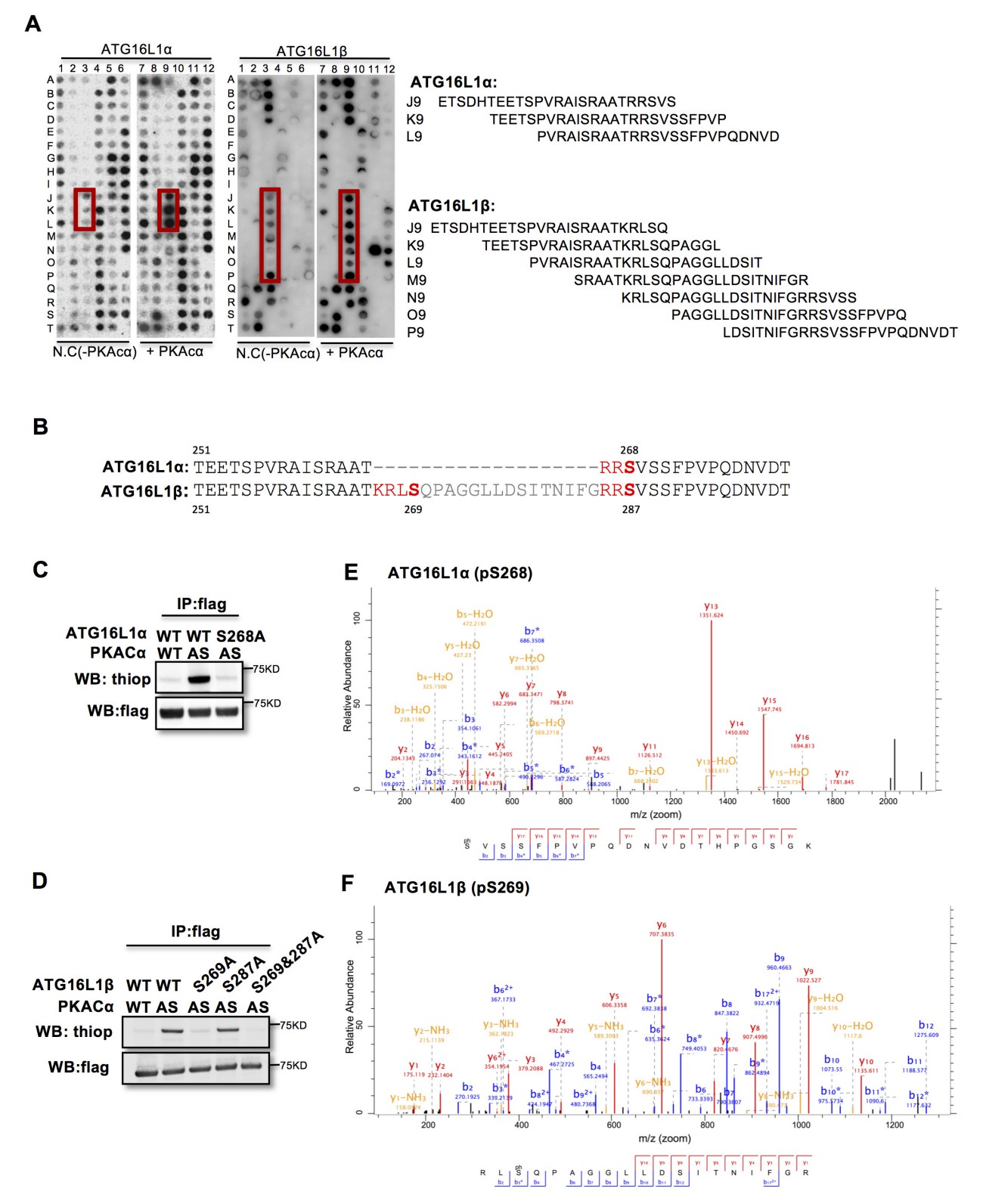

**Figure 2.** PKACα phosphorylates ATG16L1α at S268 and ATG16L1β at S269. (**A**) Peptide SPOT assay of ATG16L1 phosphorylation sites screening. (**B**) Amino acid sequence including potential PKACα phosphorylation sites in ATG16L1α and ATG16L1β according to the peptide SPOT assay result of *Figure 2A*. (**C–D**) Identification of PKACα phosphorylation site in ATG16L1α (**C**) and ATG16L1β (**D**). Analysis was performed as in *Figure 1C*. Western blots are representative of two independent experiments. (**E–F**) Flag tagged ATG16L1α and ATG16L1β were thio-phosphorylated by AS-PKACα and

*Figure 2 continued on next page*

Figure 2 continued

purified twice using M2 beads and thioP antibody coupled beads, followed by mass spectrometric analysis. LC-MS/MS spectra of the PKA-phosphorylated ATG16L1α tryptic peptide pSVSSFPVPQDNVDTHPGSGK and ATG16L1β tryptic peptide RLpSQPAGGLLDSITNIFGR. The results demonstrate that PKA phosphorylated ATG16L1α at S268 and phosphorylated ATG16L1β at S269.

The online version of this article includes the following figure supplement(s) for figure 2:

**Figure supplement 1.** Thiophophorylation is converted to normal phosphoryation by 1% TFA acid–promoted hydrolysis.

---

under control of endothelial specific Cdh5 promotor (*Nedvetsky et al., 2016*). Taken together, these results indicate that PKACα regulates ATG16L1 by phosphorylation-dependent degradation.

## Inhibition of autophagy partially normalizes the vascular phenotype caused by PKA-deficiency

ATG16L1 is an important component of the ATG16L1-ATG5-ATG12 protein complex, required for LC3 lipidation and autophagosome formation. Both LC3 lipidation and autophagosome formation represent essential steps in autophagy (*Kuma et al., 2002*; *Levine and Kroemer, 2008*; *Matsushita et al., 2007*; *Mizushima et al., 2003*; *Mizushima et al., 1999*). Accumulation of ATG16L1 upon PKA knock down resulted in increased levels of the positive autophagy marker LC3II whilst reducing the negative autophagy marker p62 in HUVECs (*Figure 3E*) whereas depleting ATG16L1 by 6-Bnz-cAMP mediated PKA activation resulted in decreased levels of LC3II whilst increasing p62 in HUVECs (*Figure 3F*). Since ATG16L1 protein levels in endothelial cells isolated from dnPKA mice were also increased, we hypothesized that increased autophagy in endothelial cells may contribute to the vascular phenotype in these mice. If so, inhibiting autophagy could potentially normalize vascular hypersprouting in dnPKA[iEC] mice. To test this hypothesis, we crossed the ATG5 conditional knock out mice ATG5[ECKO] (ATG5[flox/flox] mice crossed with Cdh5-CreERT2 mice) with dnPKA[iEC] mice, performed retinal staining for isolectin B4 (IB4, membrane staining) and the tip cell specific marker ESM1, and quantified the IB4 and ESM1 positive areas. No significant difference were observed between Cdh5-CreERT2 control mice and ATG5[ECKO] mice on both the vascular plexus and tip cells. However, ATG5[ECKO] partially rescued both the hyperdense vascular plexus front and the increasing tip cells in dnPKA[iEC] mice. Retinal stainings demonstrated that ATG5 deletion in endothelial cells in vivo, which shuts down ATG5-dependent autophagy, partially normalizes the hypersprouting phenotype of dnPKA[iEC] mice (*Figure 4A–4F*). Autophagy inhibitor chloroquine (CQ) treatment confirmed that autophagy inhibition can partially rescue both the hyperdense vascular plexus front and the increasing tip cells in dnPKA[iEC] mice (*Figure 4—figure supplement 1*) . Further more, although the ratio of proliferating endothelial cells was not significant different in retinas of the four groups of mice (Cdh5-CreERT2 control; ATG5[ECKO];dnPKA[iEC]; and dnPKA[iEC] ATG5[ECKO]), the total number of endothelial cells and proliferating endothelial cells was deceased in dnPKA[iEC] ATG5[ECKO] compared to dnPKA[iEC] mouse retinas (*Figure 5A–5D*). Also the low levels of apoptotic endothelial cells showed no significant differences (*Figure 5E–5F*), suggesting that neither the rate of proliferation nor the frequency of apoptosis are drivers of PKA and ATG5 dependent vascular density and sprouting phenotypes. ATG5 deletion in endothelial cells instead appears to normalize the hypersprouting phenotype of dnPKA[iEC] mice by reducing the number of endothelial tip cells cells as well as the total number of proliferating endothelial cells. Although speculative at this time, a shorter cell cycle or more rounds of cycling per cell in the case of increased autophagy would for example explain the increased number of cells with unchanged ratio of proliferating endothelial cells. Altogether our results suggest that PKA regulates the switch from sprouting to stabilization of nascent vascular plexuses by limiting endothelial autophagy levels via ATG16L1 degradation.

## Discussion

Our chemical genetic screen and biochemical analysis identified ATG16L1 as a novel target of PKA activity in endothelial cells. The combined results demonstrate that PKA activity inhibits autophagy in cultured human umbilical vein endothelial cells (HUVEC) via the phosphorylation of ATG16L1,

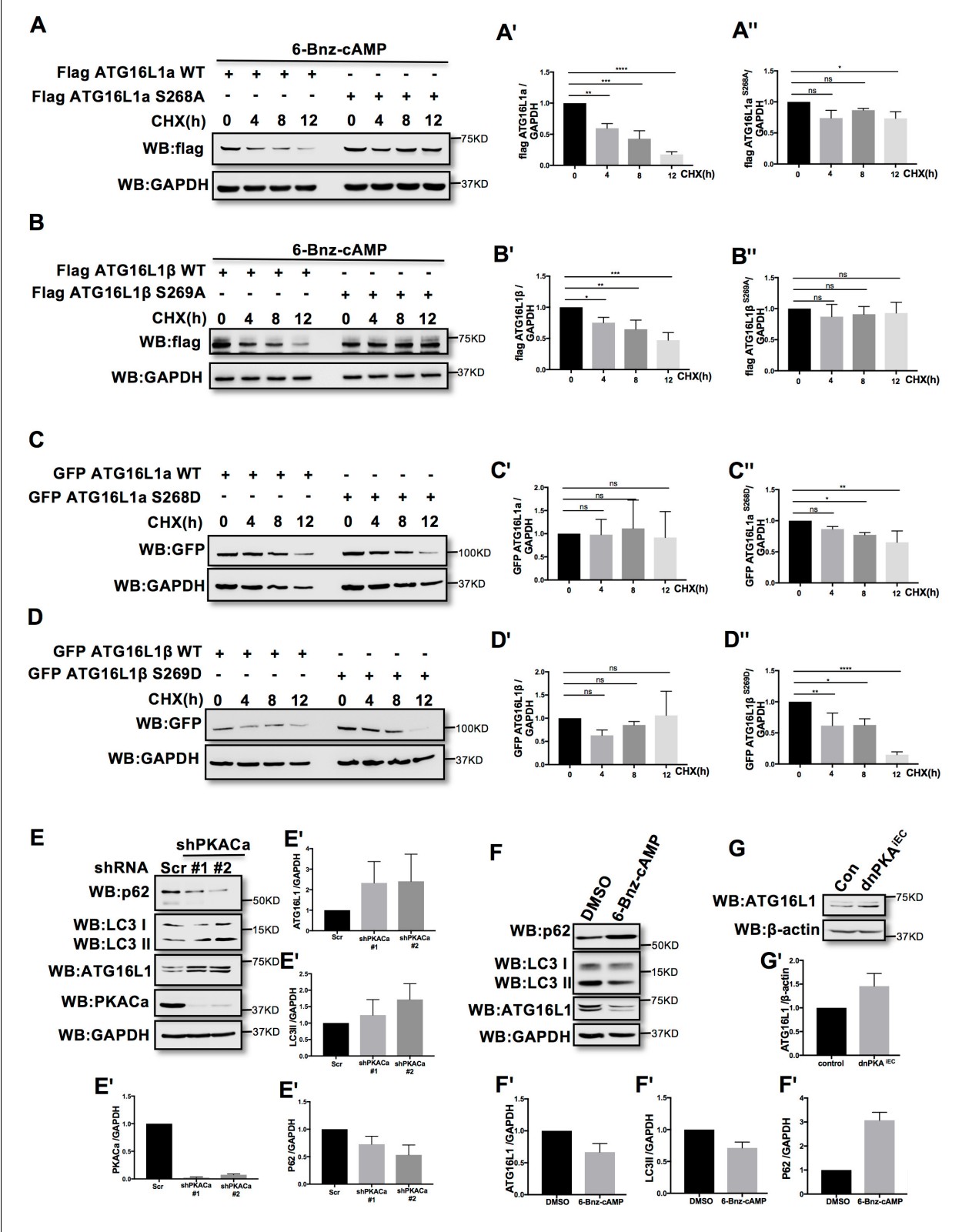

**Figure 3.** PKACα mediated phosphorylation of ATG16L1 facilitates its degradation whereas PKA deficiency stabilizes ATG16L1. (A–A′′) HUVECs infected with Flag ATG16L1α (WT or S268A) were treated with 250 μM 6-bnz-cAMP and 20 μg/ml CHX at the time points indicated when they reached confluence (A). Quantifications of Flag ATG16L1α WT (A′) and S268A (A′′) expression. (B–B′′) HUVECs infected with Flag ATG16L1β (WT or S269A) were treated with 250 μM 6-bnz-cAMP and 20 μg/ml CHX at the time points indicated when they reached confluence (B). Quantifications of Flag ATG16L1β

*Figure 3 continued on next page*

Figure 3 continued

WT (**B'**) and S269A (**B''**) expression. (**C–C''**) HUVECs infected with GFP ATG16L1α (WT or S268D) were treated with 20 µg/ml CHX at the time points indicated when they reached confluence (**C**). Quantifications of GFP ATG16L1α WT (**C'**) and S268D (**C''**) expression. (**D–D''**) HUVECs infected with GFP ATG16L1β (WT or S269D) were treated with 20 µg/ml CHX at the time points indicated when they reached confluence (**D**). Quantifications of GFP ATG16L1β WT (**D'**) and S269D (**D''**) expression. (**E–E'**) HUVECs infected with shRNA (scramble or shPKACα) virus were lysed in RIPA buffer and proteins were analyzed by western blot using indicated antibodies (**E**). Quantifications of indicated protein expression (**E'**). (**F–F'**) HUVECs treated with DMSO (control) and 500 µM 6-bnz-cAMP were lysed in RIPA buffer and proteins were analyzed by western blot using indicated antibodies (**F**). Quantifications of indicated protein expression (**F'**). (**G–G'**) Endothelial cells isolated from mice (wild type or dnPKA$^{iEC}$) were lysed in RIPA buffer and ATG16L1 protein was analyzed by western blot (**G**). Quantifications of indicated ATG16L1 expression (**G'**). Data present the mean ± SD of 3 independent experiments. *P<0,05; **P<0,01; ***P<0,001; ****P<0,0001.

The online version of this article includes the following source data for figure 3:

**Source data 1.** Values for quantification of indicated protein expression in *Figure 3A, B, C, D, E, F and G*.

which accelerates its degradation. In cultured bovine aortic endothelial cells, induction of autophagy by overexpression of ATG5 has been shown to promote in vitro vascular tubulogenesis, whereas ATG5 silencing suppressed this morphogenic behavior (*Du et al., 2012*). In mice, inhibition of autophagy by bafilomycin, or genetic beclin heterozygosity as well as ATG5 knockout impairs angiogenesis post myocardial infarction, whereas the angiogenic factor AGGF1 enhances therapeutic angiogenesis through JNK-mediated stimulation of endothelial autophagy (*Lu et al., 2016*). Angiogenesis during tissue regeneration in a burn wound model also relied on induction of endothelial autophagy, by driving AMPK/AKT/mTOR signaling (*Liang et al., 2018*), together suggesting that autophagy regulation may represent a critical determinant of the extent of vascular sprouting. The identification of ATG16L1 as a direct target of PKA therefore raises the hypothesis that the dramatic hypersprouting phenotype in dnPKA$^{iEC}$ mouse retinas deficient in endothelial PKA activity may result from exuberant endothelial activation of autophagy. Both genetic endothelial inactivation of ATG5 and chemical inhibition of autophagy partially normalized the hypersprouting phenotype in dnPKA$^{iEC}$ mice, suggesting that indeed the activation of autophagy in dnPKA$^{iEC}$ mice contributes to vascular hypersprouting. However, the failure to fully normalize vascular patterning by autophagy inhibition indicates that additional PKA targets and mechanisms may be involved. Our mass spectrometry analysis identified a list of presumptive endothelial PKA substrates, which will potentially also be involved in angiogenesis. For example, we identified RAPGEF2 as a candidate target, deficiency of which causes embryonic lethality at E11.5 due to yolk sac vascular defects (*Satyanarayana et al., 2010*), very similar to the yolk sac phenotype in dnPKA$^{iEC}$ embryos (*Nedvetsky et al., 2016*). Similarly the potential target Rock2 has a well known role in regulating endothelial functions in angiogenesis (*Liu et al., 2018*; *Montalvo et al., 2013*; *Seto et al., 2016*; *Shimizu et al., 2013*). Further studies will need to validate all the listed targets and establish which of these exert critical endothelial functions, and under what conditions.

An alternative explanation for the partial rescue of the dnPKA$^{iEC}$ phenotype by inhibition of autophagy could lie in additional functions of ATG16L1 itself. Although ATG16L1 plays an essential role in autophagy, and is part of a larger protein complex ATG16L1-ATG5-ATG12 that is necessary for autophagy, ATG16L1 is also involved in the production of inflammatory cytokines IL-1β and IL-18 and exerts anti-inflammatory functions during intestinal inflammation (*Cadwell et al., 2008*; *Diamanti et al., 2017*; *Saitoh et al., 2008*; *Sorbara et al., 2013*). IL-1β promotes angiogenesis by activating VEGF production during tumor progression (*Carmi et al., 2013*; *Voronov et al., 2003*), while IL-18 suppresses angiogenesis in cancer (*Cao et al., 1999*; *Xing et al., 2016*; *Yang et al., 2010*). How ATG16L1 regulates angiogenesis through inflammatory cytokines and whether this regulation operates downstream of PKA activity in vivo requires further investigation.

Intriguingly, our rescue experiments show that in wild type mice, inhibition of autophagy has no significant effect on developmental retinal angiogenesis. This could indicate that autophagy in developmental angiogenesis, unlike in pathological angiogenesis and post-ischemic tissue responses, is not very active.

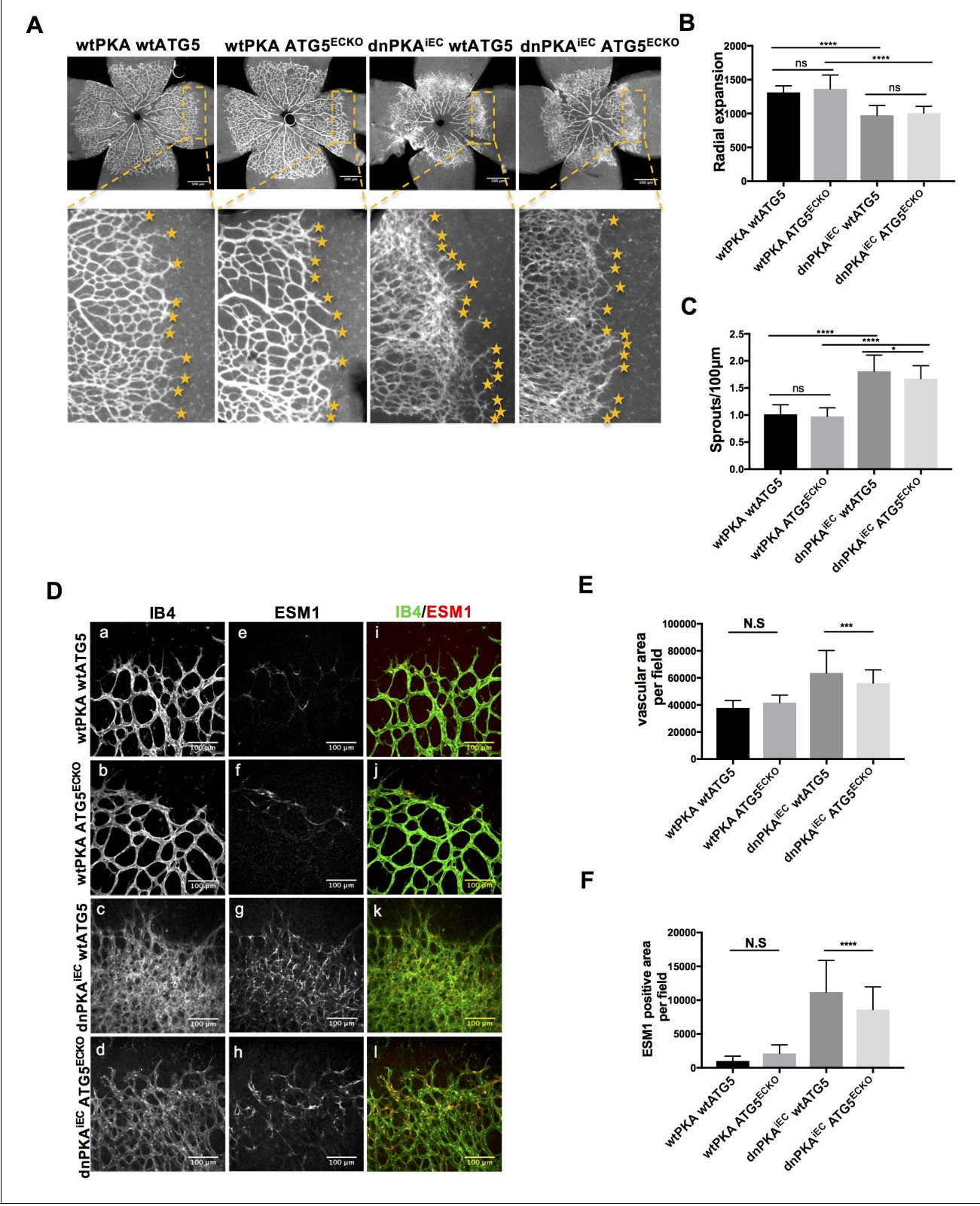

**Figure 4.** Autophagy inhibition partially rescues retinal vascular hypersprouting caused by PKA deficiency. (A–F) Mice were injected with tamoxifen from P1 to P3, then retinas were collected at P6. Isolectin B4 and ESM1 staining of P6 retinas isolated from wtPKA with wtATG5 or ATG5$^{ECKO}$ mice and dnPKA$^{iEC}$ with wtATG5 or ATG5$^{ECKO}$ mice. Representative images are shown (A,D). Quantifications of radial expansion (B), sprouts per 100μm (C),
*Figure 4 continued on next page*

*Figure 4 continued*

vascular area (**E**) and ESM1 positive area (**F**) per field of retinal fronts. 8-10 retinas were measured for each group, *P<0,05; **P<0,01; ***P<0,001; ****P<0,0001.

The online version of this article includes the following source data and figure supplement(s) for figure 4:

**Source data 1.** Values for quantification of radial expansion (*Figure 4B*), sprouts per 100 µm (*Figure 4C*), vascular area (*Figure 4E*) and ESM1 positive area (*Figure 4F*) per field of retinal fronts.

**Figure supplement 1.** Autophagy inhibition partially rescues retinal vascular hypersprouting caused by PKA deficiency.

**Figure supplement 1—source data 1.** Values for vascular area (*Figure 4—figure supplement 1C*) and ESM1 positive area (*Figure 4—figure supplement 1D*) per field of retinal fronts.

Although, to our knowledge, this is the first identification of ATG16L1 as a PKA target and the first indication that PKA regulates autophagy in endothelial cells, PKA has previously been identified as regulator of autophagy. For example, PKA reduces autophagy through phosphorylation of ATG13 in *Saccharomyces cerevisiae* (*Hundsrucker et al., 2006*), and through phosphorylation of LC3 in neurons (*Cherra et al., 2010*). In our research, ATG16L1 was identified as a novel direct PKA substrate in endothelial cells, but not ATG13 or LC3. Mechanistically, the phosphorylation of ATG16L1 by PKA accelerates its degradation, and consequently decreases autophagy levels in endothelial cells. The finding of different components of the autophagy pathway as targets of PKA identified in yeast and various vertebrate cell populations raises the intriguing possibility that although the principle regulatory logic of PKA in autophagy is conserved, different protein targets mediate this effect in different cells or organisms. In addition, or alternatively, this regulation carries multiple levels of redundancy, and the individual studies simply identify the most prevalent targets within the respective cell types. The fact that also ATG16L1 comes in two splice variants that are both targeted by PKA in endothelial cells lends some strength to this idea.

Interestingly, ATG16L1 can itself be regulated by multiple phosphorylation events by distinct kinases, with opposing effects on protein stability and autophagy. ATG16L1 can be phosphorylated at Ser139 by CSNK2 and this phosphorylation enhances its interaction with the ATG12-ATG5 conjugate (*Song et al., 2015*). IKKα promotes ATG16L1 stabilization by phosphorylation at Ser278 (*Diamanti et al., 2017*). In addition, phospho-Ser278 has similar functions as phospho-Thr300, since both phospho-mutants ATG16L1$^{S278A}$ and ATG16L1$^{T300A}$ accelerate ATG16L1 degradation by enhancing caspase three mediated ATG16L1 cleavage (*Diamanti et al., 2017*; *Murthy et al., 2014*). In contrast, our finding suggest that the PKA target sites Ser268 in ATG16L1α (or Ser269 in ATG16L1β) work in the opposite way of Ser278 and Thr300; ATG16L1α$^{S268A}$ (and ATG16L1β$^{S269A}$) are more stable than ATG16L1$^{WT}$. Furthermore, PKA deficiency also stabilizes ATG16L1 in endothelial cells in vivo and in vitro. Taken together, it appears that the different phosphorylation sites of ATG16L1 play different roles in fine tuning protein stability under the influence of alternative upstream kinases, and thereby adapt autophagy levels. Given the increasing insights into the role of autophagy in cell and tissue homeostasis and in disease, it will be of great interest to investigate whether the newly identified regulation by PKA extends beyond developmental angiogenesis into pathomechanisms associated with endothelial dysfunction.

Finally, on a technical note, the chemical genetics approach developed by Shokat and colleagues (*Alaimo et al., 2001*; *Allen et al., 2005*; *Allen et al., 2007*) has successfully been used in other cell types, but to our knowledge, this is the first report on direct endothelial PKA targets. Our initial attempts using published cell lysate conditions based on RIPA buffer however failed to identify differences in thiophosphorylation when comparing AS-PKA expressing cells to WT-PKA expressing cells. Our buffer optimization revealed that RIPA buffer limits the activity of PKA, whereas our new kinase lysis buffer (see Materials and methods) allows effective substrate phosphorylation, thus giving rise to strong signals in AS-PKA samples. This optimization will hopefully be valuable for researchers aiming to utilize this approach for additional chemical genetic kinase substrate screens in the future.

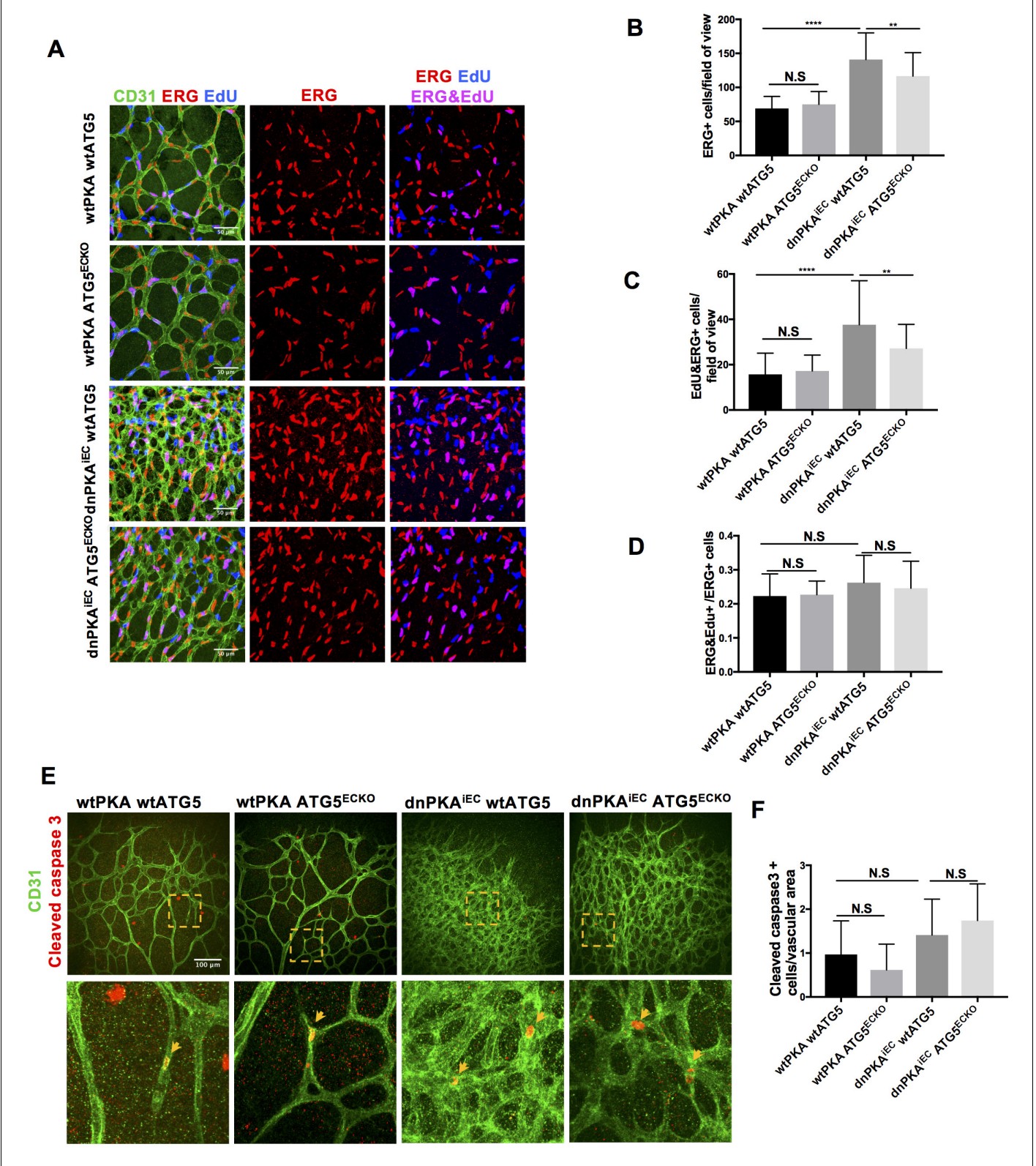

**Figure 5.** Autophagy inhibition partially rescues retinal vascular hypersprouting caused by PKA deficiency through reducing endothelial cell number but not the ratio of proliferation endothelial cells and apoptosis of endothelial cells. (A–D) Mice were injected with tamoxifen from P1 to P3, 50 μl 1 mg/ml EdU were I.P injected 2 hr before retinas collecting at P6. CD31 and ERG staining of P6 retinas isolated from wtPKA with wtATG5 or ATG5$^{ECKO}$ mice and dnPKA$^{iEC}$ with wtATG5 or ATG5$^{ECKO}$ mice followed by EdU Click-iT 647 dye labeling. Representative images are shown (**A**). Quantifications of

*Figure 5 continued on next page*

*Figure 5 continued*

endothelial cells (ERG positive cells) (B), proliferating endothelial cells (EdU and ERG positive cells) (C) and ratio of proliferating endothelial cells (EdU and ERG positive cells/ERG positive cells) (D). 6–9 retinas were measured for each group, *p<0,05; **p<0,01; ***p<0,001; ****p<0,0001. (E–F) Mice were injected with tamoxifen from P1 to P3, then retinas were collected at P6. CD31 and Cleaved caspase 3 staining of P6 retinas isolated from wtPKA with wtATG5 or ATG5^ECKO mice and dnPKA^iEC with wtATG5 or ATG5^ECKO mice. Representative images are shown (E). Quantifications of endothelial apoptosis (F). 6–8 retinas were measured for each group.

The online version of this article includes the following source data for figure 5:

**Source data 1.** Values for quantification of endothelial cells (ERG positive cells) (*Figure 5B*), proliferating endothelial cells (EdU and ERG positive cells) (*figure 5C*), ratio of proliferating endothelial cells (EdU and ERG positive cells/ERG positive cells) (*Figure 5D*) and endothelial apoptosis (*Figure 5F*).

# Materials and methods

## Key resources table

| Reagent type (species) or resource | Designation | Source or reference | Identifiers | Additional information |
|---|---|---|---|---|
| Strain, strain background (*Mus musculus*) | Prk ar1α^tm2Gsm | PMID: 21533282 | | |
| Strain, strain background (*Mus musculus*) | Tg(Cdh 5-cre/ERT2)^1Rha | MGI:3848980 | | |
| Genetic reagent (*Mus musculus*) | ATG5^flox/flox | PMID: 16625204 | | |
| Cell line (human) | HUVEC | PromoCell and freshly isolated cells | | |
| Cell line (human) | HEK293T | ATCC | | |
| Transfected construct (Mus) | pRRL.CMV.flag-PKACα | This paper | | |
| Transfected construct (Mus) | pRRL.CMV.flag-PKACαM120G | This paper | | |
| Transfected construct (human) | pECE-M2-PPP1R12A | Addgene: 31658 | | |
| Transfected construct (human) | EGFPC1-huNF ATc1EE-WT | Addgene: 24219 | | |
| Transfected construct (human) | pDESTmycDDX17, pRRL.CMV.f lag-DDX17 | Addgene: 19876 This paper | | |
| Transfected construct (human) | pMRX-IP/SECFP-h ATG16L1 pRRL.CMV. flag- ATG16L1α | Addgene: 58994 This paper | | |
| Transfected construct (human) | pRRL.CMV.flag- ATG16L1β | This paper | | |
| Transfected construct (human) | pRRL.CMV. GFP- ATG5 | This paper | | |

*Continued on next page*

*Continued*

| Reagent type (species) or resource | Designation | Source or reference | Identifiers | Additional information |
|---|---|---|---|---|
| Transfected construct (human) | pRRL.CMV.flag- ATG16L1α S268A | This paper | | |
| Transfected construct (human) | pRRL.CMV.flag- ATG16L1β S269A | This paper | | |
| Transfected construct (human) | pRRL.CMV.flag- ATG16L1β S287A | This paper | | |
| Transfected construct (human) | pRRL.CMV.flag- ATG16L1β S269A and S287A | This paper | | |
| Transfected construct (human) | pRRL.CMV.GFP- ATG16L1α | This paper | | |
| Transfected construct (human) | pRRL.CMV.GFP- ATG16L1α S268D | This paper | | |
| Transfected construct (human) | pRRL.CMV.GFP- ATG16L1β | This paper | | |
| Transfected construct (human) | pRRL.CMV.GFP- ATG16L1β S269D | This paper | | |
| Transfected construct (human) | pLKO.1-TRC cloning shRNA vector | Addgene: 10878 | | |
| Transfected construct (human) | pLKO.1-TRC shPKACα | This paper | | |
| Antibody | Anti-Thiophosphate ester antibody [51-8] | Abcam: ab92570 | | WB 1:5000 |
| Antibody | Anti-Thiophosphate ester antibody [51-8] | Abcam: ab133473 | | IP |
| Antibody | GAPDH (14C10) Rabbit mAb | cell signaling: #2118 | | WB 1:1000 |
| Antibody | Atg16L1 (D6D5) Rabbit mAb | cell signaling: #8089 | | WB 1:1000 |
| Antibody | LC3B (D11) XP Rabbit mAb | cell signaling: #3868 | | WB 1:1000 |
| Antibody | PKA C-α Antibody | cell signaling: #4782 | | WB 1:1000 |
| Antibody | SQSTM1/p62 (D5E2) Rabbit mAb | cell signaling: #8025 | | WB 1:1000 |
| Antibody | Phospho-PKA Substrate (RRXS*/T*) (100G7E) Rabbit mAb | cell signaling: #9624 | | WB 1:1000 |

*Continued on next page*

*Continued*

| Reagent type (species) or resource | Designation | Source or reference | Identifiers | Additional information |
|---|---|---|---|---|
| Antibody | GFP Tag Polyclonal Antibody | Invitrogen: A11122 | | WB 1:1000 |
| Antibody | goat anti-Actin(c-11) | Santa Cruz Biotechnology: sc-1615 | | WB 1:2000 |
| Antibody | ANTI-FLAG antibody produced in rabbit | Sigma: F7425 | | WB 1:1000 |
| Antibody | Monoclonal ANTI-FLAG M2 antibody produced in mouse | Sigma: F3165 | | WB 1:1000 |
| Antibody | Anti-rabbit IgG, HRP-linked Antibody | cell signaling: #7074 | | WB 1:2000 |
| Antibody | chicken anti-goat IgG-HRP | Santa Cruz Biotechnology: sc-516086 | | WB 1:5000 |
| Antibody | Peroxidase AffiniPure Donkey Anti-Mouse IgG (H+L) | Jackson Immuno Research: 715-035-151 | | WB 1:2000 |
| Antibody | Isolectin GS-IB4 From Griffonia simplicifolia, Alexa Fluor 488 Conjugate | Thermo Fisher: I21411 | | IF 1:100 |
| Antibody | Mouse Endocan/ ESM-1 Antibody | R and D: AF1999 | | IF 1:100 |
| Antibody | Donkey anti-Goat IgG (H+L) Cross-Adsorbed Secondary Antibody, Alexa Fluor 555 | Thermo Fisher: A21432 | | IF 1:500 |
| Antibody | Rabbit anti-ERG | Abcam: ab92513 | | IF 1:500 |
| Antibody | rabbit anti-cleaved caspase 3 | R and D: AF835 | | IF 1:200 |
| Antibody | Rat anti-CD31 | BD Pharmingen: BD553370 | | IF 1:200 IP |
| Antibody | Donkey anti-Rat IgG (H+L) Cross-Adsorbed Secondary Antibody, Alexa Fluor 488 | Thermo Fisher: A21208 | | IF 1:500 |
| Antibody | Donkey anti-Rabbit IgG (H+L) Cross-Adsorbed Secondary Antibody, Alexa Fluor 555 | Thermo Fisher: A31572 | | IF 1:500 |

*Continued on next page*

*Continued*

| Reagent type (species) or resource | Designation | Source or reference | Identifiers | Additional information |
|---|---|---|---|---|
| Commercial assay or kit | Q5 Site-Directed Mutagenesis Kit | NEB: E0554S | | |
| Commercial assay or kit | Lenti-X p24 Rapid Titer Kit | Clontech: 632200 | | |
| Commercial assay or kit | Pierce Silver Stain Kit | Thermo Fisher:24600 | | |
| Commercial assay or kit | Click-iT EdU Alexa Fluor 647 Imaging Kit | Thermo fishier:C10340 | | |
| Chemical compound, drug | X-tremeGENE HP DNA transfection reagent | Roche | | |
| Chemical compound, drug | 6-cHe-ATPγS | Biolog:C127 | | |
| Chemical compound, drug | Sp-8-CPT-cAMPS | Biolog:C012 | | |
| Chemical compound, drug | 6-bnz-cAMP | Biolog:C009 | | |
| Software, algorithm | image J | image J | | |
| Software, algorithm | GraphPad Prism 7 | GraphPad Prism 7 | | |
| Other | Recombinant Protein G Agarose | Invitrogen | | |
| Other | sheep anti-Rat IgG-coupled Dynabeads | Invitrogen | | |

## Cell culture

Human Umbilical Vein Endothelial Cells (HUVECs) were freshly isolated (or purchased from Promocell) and cultured in Endothelial Cell Growth Medium (Ready-to-use) (Promocell), passage 3 to 5 were used for experiments. HEK293T cells were cultured in Dulbecco's Modified Eagle Medium (DMEM, Thermo Fisher) with 10% Fetal Bovine Serum (FBS, Thermo Fisher) and 50 U/ml penicillin and 50 mg/ml streptomycin (Thermo Fisher) in 5% $CO_2$ at 37°C.

## Plasmid construction

Lentivirus vector pRRLsin.PPT.CMV.flag.MCS and pRRLsin.PPT.CMV.GFP.MCS were generated by resctrictional cloning of a sequence coding flag-tag/GFP-tag into the pRRLsin.PPT.CMV.MCS vector at XbaI and XmaI restriction sites. PKACα gene from mouse, ANKRD40 and ATG5 were amplified from total RNA extracted from HUVECs and cloned into the pRRLsin.PPT.CMV.flag/GFP.MCS vector. pECE-M2-PPP1R12A wt, EGFPC1-huNFATc1EE-WT, pDESTmycDDX17, pMRX-IP/SECFP-hATG16L1 were purchased from Addgene. pDESTmycDDX17, pMRX-IP/SECFP-hATG16L1 and subcloned to vector pRRLsin.PPT.CMV.flag.MCS. PKACα M120G, ATG16L1α S268A, ATG16L1β S269A, ATG16L1β S287A, ATG16L1β S269A and S287A were generated by site-directed mutagenesis using Q5 Site-Directed Mutagenesis Kit (NEB). ATG5, ATG16L1α, ATG16L1α S268D, ATG16L1β, ATG16L1β S269D were cloned into the pRRLsin.PPT.CMV.GFP.MCS vector. pLKO.1-TRC cloning shRNA vector (addgene) was used to clone PKACa shRNA constructs targeting sequence: TAGATCTCACCAAGCGCTTTG and TCAAGGACAACTCAAACTTAT.

## Lentivirus production and infection

For lentivirus production, HEK293T cells, seeded in 150 mm dishes, were transfected with flag-tagged or GFP-tagged constructs, psPAX2 and pMD2.G using X-tremeGENE HP (Roche) as transfection reagent. Medium was changed 12–16 hr after transfection. Lentivirus-containing medium was collected in 24–48 hr afterwards and filtered through 0.45 filters. Lentivius titers were determined with Lenti-X p24 Rapid Titer Kit (Clontech). To infect HUVEC, lentivirus (MOI 20–50) and polybrene (final concentration 8 µg/ml) was added to cells for 18–22 hr, and then the cells were washed with PBS and replaced the medium with fresh EGM2.

## Protein extraction and western blot

Cells were lysed in RIPA buffer contained protease inhibitor cocktail and PhosSTOP (Roche). Protein concentrations were measured by Pierce BCA Protein Assay Kit. Samples were further diluted with SDS-loading buffer and SDS-PAGE was performed using NuPAGE 4–12% Bis-Tris Protein Gels (Invitrogen). Proteins were tranferred to nitrocellulose membrane with iBlot 2 Dry Blotting System (Thermo Fisher) or to PVDF membranes by wet blotting. Membranes were blocked with 5% nonfat milk in TBST and primary antibodies were incubated overnight at 4°C or 1.5 hr at room temperature. HRP-conjugated secondary antibodies were diluted and incubated 1 hr at room temperature. Super-Signal West Pico Chemiluminescent Substrate (Thermo Fisher) was used for imaging. Following antibodies were used: rabbit anti-thiophosphate ester (ab92570,1:5000) was from Abcam, goat anti-β-actin (sc-1615,1:2000) antibody, chicken anti-goat IgG-HRP (sc-516086,1:5000) were from Santa Cruz Biotechnology, rabbit anti-GAPDH (#2118,1:1000), rabbit anti-PKACa (#4782,1:1000), rabbit anti-ATG16L1 (#8089,1:1000), rabbit anti-p62 (#8025,1:1000), rabbit anti-LC3 (#3868,1:1000) antibodies and anti-rabbit IgG, HRP-linked antibody (#7074,1:2000) were from Cell Signaling, rabbit anti-flag (F7425,1:1000) and mouse anti-flag (F3165,1:1000) were from Sigma and rabbit anti-GFP (A11122,1:1000) was from Invitrogen. Peroxidase affinipure donkey anti-mouse IgG (715-035-151,1:2000) was from Jackson Immuno Research.

## Chemical genetic screen and validation of PKA substrates

HUVECs (6 10cm-dishes each containing $1 \times 10^6$ cells) were infected with lentivirus encoding either flag-PKACα WT (as negative control) or flag-PKACα M120G. 48 hr after infection, cells were stimulated with Sp-8-CPT-cAMPS (Biolog) for 10 min, then lysed in kinase lysis buffer (1% NP40, 142 mM NaCl, 25 mM Tris-HCl (pH 7.5), 5 mM β-glycerophosphate, 2 mM dithiothreitol, 0.1 mM $Na_3VO_4$, 10 mM MgCl2) with protease inhibitor cocktail on ice for 20 min and spun (16,000g × 10 min) to remove cell debries. 3.5 mM GTP and 350 µM 6-cHe-ATPγS (Biolog) were added to the lysates. After 30 min incubation at 30°C, 2.5 mM p-Nitrobenzyl mesylate (PNBM, Abcam) was added and the reaction was incubated for additional 2 hr at room temperature. PNBM was removed by Zeba Spin Desalting Columns, 7K MWCO (Thermo Fisher) and samples were washed with IP buffer (1% NP40, 150 mM NaCl, 50 mM Tris-HCl (pH 7.5), 0.5% sodium deoxycholate). The protein fractions were precleared by incubation with Recombinant Protein G Agarose for one hour at 4°C. The precleared samples were then incubated with anti-thiophosphate ester antibody (51-8; Abcam) coupled to Recombinant Protein G Agarose coupled gently rocking overnight at 4°C. The agarose beads were washed four times with IP buffer. 1/30 of the washed beads was boiled in SDS sample loading buffer for western blot detection or silver staining. The rest samples were used for mass spectrometry analyze.

For validation of the identified substrates, 293 T cells ($1.5 \times 10^6$ on a 6 cm dish) were transfected with 0.5 µg pRRL PKACα WT or pRRL PKACα M120G and 1.5 µg of indicated candidate substrate using X-tremeGENE HP DNA transfection reagent (Roche). 30 hr after transfection, cells were stimulated with Sp-8-CPT-cAMPS (Biolog) for 10 min, lysated and treated as described above.

## Silver staining

Silver staining was performed using Pierce Silver Stain Kit (Thermo Fisher,24600) according to the manufacturer's protocol. Briefly, the SDS-page gel was washed in ultrapure water and fixed by fixing solution (30% ethanol,10% acetic acid) for 30 min. After incubating the gel in sensitizer working solution (provided in the kit) for 1 min, silver stain enhancer (provided in the kit) was added for

another 5 min. Subsequently, the gel was incubated with developer working solution (provided in the kit) for 2–3 min, before stopping the reaction with stop solution (provided in the kit).

## Mass spectrometry to identify new PKA substrates and phosphorylation sites

For mass spectrometric analysis to identify new PKA substrates, samples were prepared by chemical genetical approach as described above, each sample was run on a stacking SDS-PAGE collecting all proteins in a single band. After coomassie blue staining, the minced gel pieces were digested with trypsin based on *Shevchenko et al. (2006)* in an automated fashion using a PAL robot (Axel Semrau/CTC Analytics). Samples were measured on an LTQ Orbitrap VELOS mass spectrometer (Thermo Fisher) connected to a Proxeon nano-LC system (Thermo Fisher). Five microliters of the sample was loaded on a nano-LC column (0.074 × 250 mm, 3 mm Reprosil C18; Dr. Maisch) and separated on a 155 min gradient (4%–76% acetonitrile) at a flow rate of 0.25 µl/min and ionized using a Proxeon ion source. Mass spectrometric acquisition was done at a resolution of 60,000 with a scan range of 200–1,700 m/z in FTMS mode selecting the top 20 peaks for collision-induced dissociation fragmentation. Tandem mass spectrometric scans were measured in ion-trap mode with an isolation width of 2 m/z and a normalized collision energy of 40. Dynamic exclusion was set to 60 s. For data analysis, the MaxQuant software package version 1.5.2.8 (*Cox and Mann, 2008*) was used. Carbamidomethylation on cysteine was set as a fixed modification and oxidized methionine, acetylated N-termini and phosphorylation as variable modifications. An FDR of 0.01 was applied for peptides and proteins and the Andromeda search (*Cox et al., 2011*) was performed using a mouse Uniprot database (August 2014). MS intensities were normalized by the MaxLFQ algorithm implemented in MaxQuant (*Cox et al., 2014*). MaxLFQ-normalized intensities among the replicates of the groups to be related were used for comparison. For downstream analysis *R* was used to calculate fold changes and t-statistics.

For mass spectrometric analysis to identify phosphorylatin sites of ATG16L1, purified thiophosphorylated ATG16L1 proteins on beads were washed 3 times with 800 µl IP buffer followed by three times washing with 800 µl digestion buffer (20 mM Tris pH 8.0, 2 mM CaCl2) and dried. The washed beads were resuspended in 150 µl digestion buffer and incubated for 4 hr with 1 µg trypsin (Promega, catnr: V5111) at 37 ˚C. Beads were removed, another 1 µg of trypsin was added and proteins were further digested overnight at 37 ˚C. Peptides were acidified with 1% TFA and purified on Omix C18 tips (Agilent, catnr. A57003100), dried and re-dissolved in 20 µl loading solvent (0.1% TFA in water/acetonitrile (98:2, v/v)). Five microliters of the peptide mixture was injected for LC-MS/MS analysis on an Ultimate 3000 RSLC nano LC (Thermo, Bremen, Germany) in-line connected to a Q Exactive mass spectrometer (Thermo). Trapping was performed at 10 µl/min for 4 min in loading solvent on a 100 µm internal diameter (I.D.)×20 mm trapping column (5 µm beads, C18 Reprosil-HD, Dr. Maisch, Germany) and the sample was loaded on a reverse-phase column (made in-house, 75 µm I.D. x 220 mm, 1.9 µm. Peptides were eluted by a linear increase from 2% to 55% solvent B (0.08% formic acid in water/acetonitrile (2:8, v/v)) over 120 min at a constant flow rate of 300 nl/min. The mass spectrometer was operated in data-dependent mode, automatically switching between MS and MS/MS acquisition. Full-scan MS spectra (400–2000 m/z) were acquired at a resolution of 70,000 in the orbitrap analyzer after accumulation to a target value of 3,000,000. The five most intense ions above a threshold value of 17,500 were isolated (window of 2.0 Th) for fragmentation at a normalized collision energy of 25% after filling the trap at a target value of 50,000 for maximum 80 ms. MS/MS spectra (200–2000 m/z) were acquired at a resolution of 17,500 in the orbitrap analyzer. Raw LC-MS/MS data files were searched against the human proteins in the Uniprot/Swiss-Prot database (database version of September 2017 containing 20,237 human sequences, downloaded from www.uniprot.org). The mass tolerance for precursor and fragment ions were set to 4.5 and 20 ppm, respectively, during the main search. Enzyme specificity was set as C-terminal to arginine and lysine, also allowing cleavage at proline bonds with a maximum of two missed cleavages. Variable modifications were set to oxidation of methionine residues, acetylation of protein N-termini, phosphorylation and thiophosphorylation of serine, threonine and tyrosine residues. The minimum score for modified peptides was set to 40. The S-lens RF level was set at 50 and we excluded precursor ions with single, unassigned and charge states above five from fragmentation selection.

## Peptide arrays, Peptide SPOT assay of ATG16L1 phosphorylation sites screening

Automatic peptide SPOT synthesis was carried out as described previously (*Hundsrucker et al., 2006*; *Hundsrucker et al., 2010*; *Maass et al., 2015*; *Stefan et al., 2007*). Fmoc-protected amino acids (Intavis) and amino-modified acid-stable cellulose membranes with PEG-spacers (Intavis) were used for peptide spots synthesis on an Intavis ResPep-SL device.

For phosphorylation of the peptides by PKA (*Maass et al., 2015*), the membranes were activated in ethanol, blocked in blocking buffer (5% milk in TBS-T: Tris-HCl, 10 mM; NaCl, 150 mM; Tween 20, 0.05%; pH 7.4) for 3 hr at room temperature, and washed twice with incubation buffer (Tris-HCl, 50 mM; MgCl$_2$, 5 mM; ATP, 100 µM). His-tagged recombinant catalytic subunits (vector pET46) were purified from *E. coli* (strain Rosetta D3) as described (*Maass et al., 2015*; *Schäfer et al., 2013*). The membranes were incubated with the recombinant protein (1 nM) in incubation buffer (1 hr, 30℃), washed three times with TBS-T, and phosphorylated serines were detected with PKA phosphosubstrate antibody directed against the consensus site RRX p(S/T) (Cell Signaling Technology, 100G7E, rabbit mAB #9624) in blocking buffer overnight at 4℃ (*Christian et al., 2011*). The membranes were washed three times with TBS-T, and a secondary horseradish peroxidase (HRP)-coupled donkey anti-rabbit antibody (#711-036-153; Jackson Immuno Research) was added (3 hr, RT). After three washs with TBS-T, an ECL system (Immobilon Western substrate, Merck Millipore) and an Odyssey FC device (Li-Cor ) was used for visualizing phosphorylated serines.

## Animal procedures

All animal experimental procedures were approved by the Institutional Animal Care and Research Advisory Committee of the University of Leuven and performed according to the European guidelines. Following mouse strains were used: Prkar1a$^{tm2Gsm}$ (*Willis et al., 2011*), Tg(Cdh5-cre/ERT2)$^{1Rha}$ (*Wang et al., 2010*) and ATG5$^{flox/flox}$ (*Hara et al., 2006*). All animals used in the experiments were of mixed N/FVB x C57/Bl6 background. For chloroquine rescue retinal angiogenesis experiment, pups were intraperitoneally injected with 50 µl of 1 mg/ml tamoxifen from postnatal day one (P1) to P3, and 100 µl of 1.25 mg/ml chloroquine or PBS from P1 to P5. Mice were euthanized at P6, and dissection and staining of the retinas were performed as described below. For ATG5 deletion in endothelial cells rescue retinal angiogenesis experiment and apoptosis assay in retinal endothelial cells, pups were intraperitoneally injected with 50 µl of 1 mg/ml tamoxifen from P1 to P3 and euthanized at P6. For proliferation assay in retinal endothelial cells, 50 µl of 1 mg/ml EdU (Thermo fishier, C10340) were intraperitoneally injected 2 hr before euthanizing the mice at P6. For endothelial cells isolation, pups were intraperitoneally injected with 75 µl of 1 mg/ml tamoxifen daily from P7 until P10 and the mice were euthanized at 8 weeks and endothelial cells were isolated as described below.

## Retinal angiogenesis assay

To analyse retinal angiogenesis, the procedures of isolation and staining of the retinas were performed as published (*Pitulescu et al., 2010*). Briefly, retinas were dissected in PBS and blocked/permeabilized in retina blocking buffer (1% BSA and 0.3% Triton X-100 in PBS) for 1–2 hr at room temperature. Alexa Fluor 488 conjugated Isolectin GS-IB4 (Invitrogen) diluted in Pblec solution (1 mM MgCl$_2$, 1 mM CaCl$_2$, 0.1 mM MnCl$_2$ and 1% Triton X-100 in PBS) was added to visualize whole-retina vasculature by incubating overnight at 4℃, followed by staining for ESM1 (primary goat anti-ESM1 antibody; R and D Systems). After mounting, images of retinas were taken using a Leica SPE confocal microscope equipped with a HC PL APO 20X/0.75 IMM CORR CS2 objective or Leica SP8 confocal microscope equipped with a HCX IRAPO L 25X/0.95 W objective. Images were taken at room temperature using Leica LAS X software and processed with image J software.

## Retinal endothelial proliferation and apoptosis assay

To perform endothlial proliferation and apoptosis assays, mouse eyes were collected at P6 and fixed for 30 min at room temperature with 4% PFA. Retinas were dissected in PBS and blocked/permeabilized in 1% BSA, 50 µg/ml digitonin in PBS, primary antibodies (rat anti-CD31,BD553370; rabbit anti-ERG,ab92513; rabbit anti-Cleaved caspase 3,AF835) were incubated overnight at 4℃ and secondary antibodies (Thermo fisher) were incubated for 2 hr at room temperature. Both antibodies were

diluted in1% BSA, 2% donkey serum, 50 µg/ml digitonin in PBS. The Click-iT Edu cell proliferation kit (C10340) was used to visualize proliferating endothelial cells. After mounting, images of retinas were taken using a Leica SPE confocal microscope equipped with a ACS APO 40X/1.15 oil CS objective or Leica SP8 confocal microscope equipped with a HCX IRAPO L 25X/0.95 W objective. Images were taken at room temperature using Leica LAS X software and processed with image J software.

### Endothelial cell isolation from liver or lung

Livers or lung lobes were collected in dry 10 cm dishes and minced finely with blades for one minute, and then incubated in 25 ml of pre-warmed Dulbecco modified Eagle medium (4.5 g/L glucose with L-glutamine) containing 2 mg/mL collagenase (Invitrogen) in 50 ml tubes, gently shaking for 45 min at 37°C. Suspensions were passed through a 70 µm cell strainer (VWR) and cells were spun down at 400 g for 8 min at 4°C. Pellets were resuspended in 10 ml Dulbecco modified Eagle medium containing 10% FBS, 50 U/ml penicillin and 50 µg/ml streptomycin, passed through 40 µm Nylon cell strainer (BD Falcon, Cat. No. 352340) and centrifuged at 400 g for 8 min at 4°C. Cells were resuspended in cold DPBS (1 ml/lung and 2 ml/liver), added to sheep anti-Rat IgG-coupled Dynabeads (Invitrogen) pre-incubated with purified Rat Anti-Mouse CD31 (BD Pharmingen) and incubated at 4°C for 20 min. The beads were separated using a magnetic particle concentrator (Dynal MPC-S; Invitrogen) and washed with cold DPBS with 0.1% BSA. This washing step was repeated five times after which cells were lysed in RIPA buffer for Western Blotting.

### Statistical analysis

Statistical analyses were performed using GraphPad Prism 7. The one-way ANOVA was used to compare more than two experimental groups.

## Acknowledgements

We are grateful to Evy Timmerman and Francis Impens for their help to identify the phosphorylation sites of ATG16L1 in MS analysis at the VIB proteomics core facility. We are grateful to professor Chantal Boulanger for providing the ATG5$^{flox/flox}$ mice. This work was supported by grants from the Fonds voor Wetenschappelijk Onderzoek (FWO) [G.0742.15N to HG]; the Vlaams Instituut voor Biotechnologie (VIB) Tech watch grant [Q3 2015 to PIN]; the Elsa-Kroener-Stiftung [2014_A26 to HG and PIN]; the European Research Consortil [ERC Consolidator grant REshape 311719 to HG]; This work was supported by grants from the German Centre for Cardiovascular Research (DZHK) partner site Berlin (81XZ100146), the Deutsche Forschungsgemeinschaft (DFG KL1415/7-1 and 394046635-SFB 1365) and the Bundesministerium für Bildung und Forschung (BMBF; 16GW0179K) to EK.

## Additional information

### Competing interests

Holger Gerhardt: Reviewing editor, *eLife*. The other authors declare that no competing interests exist.

### Funding

| Funder | Grant reference number | Author |
| --- | --- | --- |
| Europees Fonds voor Regionale Ontwikkeling | G.0742.15N | Holger Gerhardt |
| Else-Kroener Stiftung | 2014_A26 | Pavel Nedvetsky Holger Gerhardt |
| European Research Council | 311719 REshape | Holger Gerhardt |
| Deutsche Forschungsgemeinschaft | DFG KL1415/7-1 | Enno Klussmann |
| Deutsche Forschungsgemeinschaft | 394046635 - SFB 1365 | Enno Klussmann |

| Bundesministerium für Bildung und Forschung | 16GW0179K | Enno Klussmann |
| --- | --- | --- |
| Vlaams Instituut voor Biotechnologie | Tech Watch Q3 2015 | Pavel Nedvetsky<br>Holger Gerhardt |

The funders had no role in study design, data collection and interpretation, or the decision to submit the work for publication.

### Author contributions

Xiaocheng Zhao, Conceptualization, Data curation, Formal analysis, Validation, Investigation, Visualization, Methodology, Writing—original draft; Pavel Nedvetsky, Conceptualization, Data curation, Formal analysis, Supervision, Funding acquisition, Investigation, Methodology, Project administration, Writing—review and editing; Fabio Stanchi, Formal analysis, Investigation, Methodology, Writing—review and editing; Anne-Clemence Vion, Conceptualization, Resources, Data curation, Investigation, Methodology; Oliver Popp, Kerstin Zühlke, Formal analysis, Investigation, Methodology; Gunnar Dittmar, Supervision, Methodology; Enno Klussmann, Conceptualization, Funding acquisition, Investigation, Methodology; Holger Gerhardt, Conceptualization, Supervision, Funding acquisition, Project administration, Writing—review and editing

### Author ORCIDs

Xiaocheng Zhao (ID) https://orcid.org/0000-0002-4048-6813
Anne-Clemence Vion (ID) http://orcid.org/0000-0002-2788-2512
Enno Klussmann (ID) http://orcid.org/0000-0003-4004-5003
Holger Gerhardt (ID) https://orcid.org/0000-0002-3030-0384

### Ethics

Animal experimentation: All animal experimental procedures were approved by the Institutional Animal Care and Research Advisory Committee of the University of Leuven (application P249/2014) and performed according to the European guidelines.

### Decision letter and Author response

Decision letter https://doi.org/10.7554/eLife.46380.sa1
Author response https://doi.org/10.7554/eLife.46380.sa2

## Additional files

### Supplementary files

• Supplementary file 1. Array map of spot-synthesized 25-mer overlapping peptides covering the entire ATG16L1 protein.

• Transparent reporting form

### Data availability

The mass spectrometry proteomics data have been deposited to the ProteomeXchange Consortium via the PRIDE partner repository with the dataset identifier PXD012975. All data generated or analysed during this study are included in the manuscript and supporting files. Source data files have been provided for Figures 3, 4 and 5.

The following dataset was generated:

| Author(s) | Year | Dataset title | Dataset URL | Database and Identifier |
| --- | --- | --- | --- | --- |
| Dittmar G, Gerhardt H | 2019 | Endothelial PKA targets ATG16L1 to regulate angiogenesis by limiting autophagy | https://www.ebi.ac.uk/pride/archive/projects/PXD012975 | PRIDE, PXD012975 |

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
