## [Decision Letter]

Thank you for submitting your article "Endothelial PKA targets ATG16L1 to regulate angiogenesis by limiting autophagy" for consideration by *eLife*. Your article has been reviewed by three peer reviewers, including Gou Young Koh as the Reviewing Editor and Reviewer #1, and the evaluation has been overseen by Didier Stainier as the Senior Editor.

The reviewers have discussed the reviews with one another and the Reviewing Editor has drafted this decision to help you prepare a revised submission.

The comments of all three reviewers are in good agreement. While the reviewers found this work to be of high interest, they raised minor concerns about the strength of the conclusions that can be drawn at this stage. The authors are required to carefully address all comments point-by-point in a data-driven manner or with further analyses. Specifically, the present data fail to fully link the angiogenic morphogenesis with the endothelial cell autophagy. Therefore, additional work to detect the changes in the endothelial cell autophagy by performing immunoblotting and/or immunohistochemical experiments.

Reviewer #1:

Vascular sprouting is regulated positively and negatively by various signals, including the VEGFA-VEGFR2 and the Dll4-Notch signals. The authors of the present manuscript previously showed that the endothelial PKA negatively regulates vascular sprouting by suppressing tip cell formation, independently of the Notch signal (Nedvetsky et al., 2016). As a follow-up study of this seminal work, Zhao et al. identified autophagy-related-protein-16-like 1 (ATG16L1) as a substrate of endothelial PKA (Figure 1). Indeed, PKA phosphorylated ATG16L1alpha at S268 and ATGT16L1beta at S269 (Figure 2), and facilitated their protein degradation (Figure 3), which may lead to reduced endothelial autophagy. Accordingly, PKA deficiency stabilized ATG16L1, thereby increasing levels of the positive autophagy marker LC3 II, and reducing the negative autophagy marker p62 in cultured HUVECs (Figure 3). Finally, the authors showed that autophagy inhibition by chloroquine or endothelial ATG5 knockout partially rescued retinal vascular hyper-sprouting caused by PKA deficiency (Figure 4).

The overall quality of the chemical genetics screen and biochemical analyses sufficiently supports the notion that PKA mediates phosphorylation and degradation of ATG16L1. On the other hand, additional in vivo data are desirable to more firmly correlate endothelial autophagy with the control of angiogenesis. For example, detection of autophagy markers, such as LC3, in developing retinal vessels of WT and mutant mice shown in Figure 4 will strengthen the authors' claims.

Reviewer #2:

This is a well written and well organized article that uses an innovative chemical biology screen to identify new protein kinase A substrates including the ATG16L regulator of autophagy. Clear biochemical and cellular evidence is presented showing that phosphorylation of Ser268 on the ATG16La isoform and Ser 269 on the ATG16Lb isoforms drive degradation of the protein. These studies are followed up by some well-reasoned cell based studies on a variety of defined genetic backgrounds. This allows the authors to propose that endothelial PKA activity restricts active sprouting of blood vessels in the retina by reducing phosphorylation dependent role of ATG16L1 isoforms in endothelial autophagy. Overall, this is a nice report that provides a succinct message. I feel that it would be appropriate for the authors to expand on some of their findings as the current work could be considered a minimal publishable unit. Although the data is of high quality and necessary rigor some of the work could be presented in a more complete and convincing manner. Specific criticisms are listed below.

1) The authors convincingly show that phosphorylation of Ser268 on the ATG16La isoform and Ser 269 on the ATG16Lb isoforms drive degradation of the protein. All of this phosphorylation is believed to proceed through PKA. However, it seems very likely that other kinases can modify this basophilic site. Experiments should be incorporated that address this issue.

2) The workflow for experiments in Figure 1 is excellent, but the data could be more convincingly presented. For example, Figures 1B and 1C should be enlarged to show the data more clearly and molecular weight markers should be indicated on each gel.

3) Are their physiological agonists that can be used to recapitulate the experiments in Figure 3?

4) What are the additional spots on the peptide arrays in Figure 2A? Focusing in on the relevant information may be advisable.

5) The Mass spec traces in Figures 2E and 2F are hard to see. Need to be enlarged and enhanced to emphasize the important data.

6) What happens in the experiments in Figure 3 when phosphosite mimics (Asp or Glu) are introduced to positions 268 on ATG16La and Ser 269 on ATG16Lb. Are these forms more labile or are refractory to the process?

7) The retina images in Supplementary Figure 3 provide an appreciated context to the study and should be incorporated into the body of the text.

Reviewer #3:

The study by Zhao et al. provides mechanistic insights into regulation of angiogenesis by cAMP-dependent protein kinase A (PKA) through inhibition of endothelial autophagy. The authors employed and optimized a chemical genetic screen to find a substrate of PKA in human umbilical vein endothelial cells (HUVECs), and identified ATG16L1 (Autophagy related 16 Like 1) as a novel target of endothelial PKA. They demonstrated that PKA regulates autophagy in ECs through phosphorylation-dependent degradation of ATG16L1. Furthermore, they showed that genetic and pharmacological inhibition of autophagy partially rescue the hyper-sprouting vascular phenotype of PKA-deficient mice. Overall, this study unravels a role of endothelial PKA in regulation of autophagy in ECs and introduces a valuable tool for identifying kinase substrates in ECs. Therefore, I consider this work suitable for publication in *eLife* after addressing following points.

1) The authors should provide explanations on Prkar1a gene and meaning of 'dnPKA' for readers who are not familiar with these molecules, as they did in a previous study (Nedvetsky et al., 2016).

"To study the role of PKA in vascular development we inhibited PKA in endothelial cells using knock-in mice carrying a single floxed dominant-negative Prkar1a allele (dnPKA; Figure 1A) knocked into the genomic Prkar1a locus allowing tissue-specific inhibition of PKA (Willis et al., 2011). The regulatory PRKAR1A subunit of PKA is an endogenous inhibitor of PKA, which binds and keeps the catalytic subunits inactive under low cAMP levels (Kumon et al., 1970; Tao et al., 1970)."

2) In Figure 3G, the authors showed that knockdown of PKA results in accumulation of ATG16L1 protein, thereby increasing a positive autophagy marker, LC3II and reducing a negative autophagy marker, p62 in HUVECs. They further showed increased ATG16L1 protein in isolated ECs from dnPKAiEC mice in Figure 3H. However, the authors should provide evidences of changes in vascular autophagy in dnPKAiEC and dnPKAiEC; ATG5ECKO mice, such as results of LC3 or p62 immunoblot or immunostaining.

3) In Figure 4, the authors showed vascular hypersprouting in PKA-deficient mice and partial rescue of this phenotype by autophagy inhibition, comparing vascular area and ESM1-positive area of retina from each groups. Detailed analyses and quantitation of vascular features such as radial lengths, number of sprouts, and ratio of proliferating ECs are required.

4) In Figure 4, what is an underlying mechanism for normalization of hypersprouting vascular phenotype of PKA-deficient mice by autophagy inhibition? Does inhibition of autophagy decrease proliferation, or induce apoptosis of ECs?

---

## [Author Response]

Reviewer #2:This is a well written and well organized article that uses an innovative chemical biology screen to identify new protein kinase A substrates including the ATG16L regulator of autophagy. Clear biochemical and cellular evidence is presented showing that phosphorylation of Ser268 on the ATG16La isoform and Ser 269 on the ATG16Lb isoforms drive degradation of the protein. These studies are followed up by some well-reasoned cell based studies on a variety of defined genetic backgrounds. This allows the authors to propose that endothelial PKA activity restricts active sprouting of blood vessels in the retina by reducing phosphorylation dependent role of ATG16L1 isoforms in endothelial autophagy. Overall, this is a nice report that provides a succinct message. I feel that it would be appropriate for the authors to expand on some of their findings as the current work could be considered a minimal publishable unit. Although the data is of high quality and necessary rigor some of the work could be presented in a more complete and convincing manner. Specific criticisms are listed below.

We thank this reviewer for appreciating the value of our work and have now provided additional data and improved presentation.

1) The authors convincingly show that phosphorylation of Ser268 on the ATG16La isoform and Ser 269 on the ATG16Lb isoforms drive degradation of the protein. All of this phosphorylation is believed to proceed through PKA. However, it seems very likely that other kinases can modify this basophilic site. Experiments should be incorporated that address this issue.

This is indeed an interesting point that warrants attention. In Group-based Prediction system v3.0 silico analysis predicted 5 most likely kinases (PKG, PRKD2,MSK, AurA,PAK1) that may be able to phosphorylate Ser268 on the ATG16La isoform and Ser 269 on the ATG16Lb isoform. We used the peptide SPOT ARRAY ASSAY to test whether the two kinases with highest predicted scores, PAK1 and PRKD2, phosphorylate identified PKA sites on ATG16L1.

**Author response image 1. respfig1:** Spot array assay testing additional kinases predicted to phosphorylate the identified PKA sites on ATG16L1.

The peptide TEETAPVRAISRAATRRSVSSFPVP was spot-synthesised in triplets and phosphorylated in vitro by the indicated kinases (PKAca, PRKD2 or PAK1). As negative controls, the kinases omitted (-). Green spots in the upper lane and black in the lower lane show peptide phosphorylation

The result shows that PRKD2 but not PAK1 can phosphorylate site Ser268 on the peptide. This result suggests that indeed, as mentioned by the reviewer, other kinases can in principle modify this basophilic site. However, it also demonstrates a degree of specificity already present in the in vitro peptide assay. Whether this site can also be phosphorylated by PRKD2 in intact cells and in importantly in endothelial cells will need to be investigated in the future. Given that ATG16L1 stability in isolated ECs from control and dnPKA^iEC^ mice shows dependency on PKA activity, and deficiency of PKA phosphorylation on ATG16L1 could not be compensated by other kinases in endothelial cells, our present work would support the idea that PKA is a major regulator of ATG16L1 in endothelial cells with limited effects of other kinases in the biology investigated.

2) The workflow for experiments in Figure 1 is excellent, but the data could be more convincingly presented. For example, Figures 1B and 1C should be enlarged to show the data more clearly and molecular weight markers should be indicated on each gel.

We agree and have enlarged the pictures and added molecular weight markers.

3) Are their physiological agonists that can be used to recapitulate the experiments in Figure 3?

This is an interesting point, but complicated by the fact that physiological agonists seem to activate both Epac and PKA. For example, we tested the physiological agonist isoproterenol hydrochloride in HUVECs. As shown in Author response image 2, isoproterenol activated autophagy in HUVECs.

**Author response image 2. respfig2:** HUVECs treated with PBS(control) and 10 µM isopropanol were lysed in RIPA buffer and proteins were analyzed by western blot using indicated antibodies.

This result is consistent with a previously published paper entitled “Exchange protein directly activated by cAMP 1 promotes autophagy during cardiomyocyte hypertrophy” (Laurent AC, Cardiovasc Res. 2015 Jan 1;105(1):55-64). The authors demonstrate that isoproterenol increase autophagy in cardiomyocytes through Epac.

At least in their assay and cell type, Epac appears to be the dominant effector of isoproterenol. Despite our efforts, we could not find a physiological agonist that would only activate PKA but not Epac. Nevertheless, in our hands, the PKA specific activator 6-Bnz-cAMP, as shown in Figure 3F inhibits autophagy in HUVECs.

4) What are the additional spots on the peptide arrays in Figure 2A? Focusing in on the relevant information may be advisable.

The red boxed dots correspond to overlapping amino acid sequences that show significant differences in intensity between negative control and PKA catalytic subunit α. Other single dots cannot be considered reliable as identical core sequences are present in adjacent dots showing no signal. Dots that show equal signal on both arrays must be considered background. We nevertheless choose to present the entire array to allow the reader to appreciate the specificity of the results. Full map details are now provided as Figure 2—figure supplement 1.

5) The Mass spec traces in Figures 2E and 2F are hard to see. Need to be enlarged and enhanced to emphasize the important data.

We agree and now provide higher resolution to appreciate the details.

6) What happens in the experiments in Figure 3 when phosphosite mimics (Asp or Glu) are introduced to positions 268 on ATG16La and Ser 269 on ATG16Lb. Are these forms more labile or are refractory to the process?

The results in Figure 3C and 3D show that the GFP tagged phosphomimetic site mutants Ser268D on ATG16La and Ser269D on ATG16Lb are more labile.

7) The retina images in Supplementary Figure 3 provide an appreciated context to the study and should be incorporated into the body of the text.

We agree and have rearranged the figures and figure legends. Please see Figure 4 and Figure 4—figure supplement 1 in the revised version.

Reviewer #3:The study by Zhao et al. provides mechanistic insights into regulation of angiogenesis by cAMP-dependent protein kinase A (PKA) through inhibition of endothelial autophagy. The authors employed and optimized a chemical genetic screen to find a substrate of PKA in human umbilical vein endothelial cells (HUVECs), and identified ATG16L1 (Autophagy related 16 Like 1) as a novel target of endothelial PKA. They demonstrated that PKA regulates autophagy in ECs through phosphorylation-dependent degradation of ATG16L1. Furthermore, they showed that genetic and pharmacological inhibition of autophagy partially rescue the hyper-sprouting vascular phenotype of PKA-deficient mice. Overall, this study unravels a role of endothelial PKA in regulation of autophagy in ECs and introduces a valuable tool for identifying kinase substrates in ECs. Therefore, I consider this work suitable for publication in eLife after addressing following points.1) The authors should provide explanations on Prkar1a gene and meaning of 'dnPKA' for readers who are not familiar with these molecules, as they did in a previous study (Nedvetsky et al., 2016)."To study the role of PKA in vascular development we inhibited PKA in endothelial cells using knock-in mice carrying a single floxed dominant-negative Prkar1a allele (dnPKA; Figure 1A) knocked into the genomic Prkar1a locus allowing tissue-specific inhibition of PKA (Willis et al., 2011). The regulatory PRKAR1A subunit of PKA is an endogenous inhibitor of PKA, which binds and keeps the catalytic subunits inactive under low cAMP levels (Kumon et al., 1970; Tao et al., 1970)."

We greatly appreciate this comment and have now provided more details to explain the meaning of dnPKA in the text as below:

“DnPKA^iEC^ is the short denomination for Prkar1aTg/+ mice carrying a single floxed dominant-negative Prkar1a allele, (the regulatory subunit Prkar1a of PKA is an endogenous inhibitor of PKA) crossed with Cdh5-CreERT2 mice expressing tamoxifen inducible Cre recombinase under control of endothelial specific Cdh5 promotor (Nedvetsky et al., 2016).”

2) In Figure 3G, the authors showed that knockdown of PKA results in accumulation of ATG16L1 protein, thereby increasing a positive autophagy marker, LC3II and reducing a negative autophagy marker, p62 in HUVECs. They further showed increased ATG16L1 protein in isolated ECs from dnPKA^iEC^ mice in Figure 3H. However, the authors should provide evidences of changes in vascular autophagy in dnPKA^iEC^ and dnPKA^iEC^; ATG5^ECKO^ mice, such as results of LC3 or p62 immunoblot or immunostaining.

We appreciate the wish to see evidence for altered endothelial autophagy levels in vivo. Unfortunately, this proved despite all our efforts not to be feasible. Having worked next door to autophagy experts (Sharon Tooze) for many years, I was aware of the tricky nature of the question. Autophagosomes are very small structures and staining for LC3 II as marker for autophagosome processing notoriously difficult in vivo. Reliable and quantifiable results are only achievable on isolated protein. We tested several of the best antibodies and could not achieve reliable staining on retinas. Efforts to isolate the endothelial cells demonstrated that we obtain too little protein from endothelial cells (a P5 mouse retina contains roughly 20 to 30 thousand endothelial cells, or which only 5% are of relevance because they are in the very front. Even when pooling from several mice, we do not obtain sufficient material for reliable immunoblotting. Given that we require up to 5 different alleles in our transgenic compound mice for this question, even with extensive breeding we do not reach the numbers to perform this experiment. Now that we tried, and are several month past resubmission timeline, we have to conclude that we will not be able to provide the in vivo evidence. Nevertheless, we hope the reviewer will agree that the mechanistic evidence in endothelial cells is convincingly demonstrating that PKA through phosphorylation mediated regulation of ATG16L1 stability regulates endothelial autophagy levels. How exactly autophagy levels drive and fine tune endothelial behavior in vivo will need to be part of future investigations.

3) In Figure 4, the authors showed vascular hypersprouting in PKA-deficient mice and partial rescue of this phenotype by autophagy inhibition, comparing vascular area and ESM1-positive area of retina from each groups. Detailed analyses and quantitation of vascular features such as radial lengths, number of sprouts, and ratio of proliferating ECs are required.

We agree and have collected more retinas, performed additional stainings, optimized segmentation and quantification and now provide additional quantification of radial lengths, and number of sprouts/100µm (see Figure 4). We also performed EdU incorporation assay, staining and segmentation to quantify the ratio of proliferating ECs (see Figure 5D).

4) In Figure 4, what is an underlying mechanism for normalization of hypersprouting vascular phenotype of PKA-deficient mice by autophagy inhibition? Does inhibition of autophagy decrease proliferation, or induce apoptosis of ECs?

We have now performed the proliferation and apoptosis assay in mouse retinas (see Figure 5). The results show that inhibition of autophagy decreases proliferation but has no significant effects on apoptosis. It appears that the combined reduction of tip cells and decrease in total number of proliferating endothelial cells can provide a cellular mechanism for normalization.